# The effect of a ketogenic diet and synergy with rapamycin in a mouse model of breast cancer

Yiyu Zou[1], Susan Fineberg[2], Alexander Pearlman[1], Richard D. Feinman[3], Eugene J. Fine[1,2]*

1 Albert Einstein College of Medicine, Bronx, NY, United States of America, 2 Montefiore Medical Center, Bronx, NY, United States of America, 3 SUNY Downstate Health Sciences Center, Brooklyn, NY, United States of America

* eugene.fine@einsteinmed.org

**Data Availability Statement:** All relevant data are within the manuscript and its Supporting Information files.

**Funding:** Generous funding for this project was provided by ST Balchug, a commercial company

## Abstract

### Background

The effects of diet in cancer, in general, and breast cancer in particular, are not well understood. Insulin inhibition in ketogenic, high fat diets, modulate downstream signaling molecules and are postulated to have therapeutic benefits. Obesity and diabetes have been associated with higher incidence of breast cancer. Addition of anti-cancer drugs together with diet is also not well studied.

### Methods

Two diets, one ketogenic, the other standard mouse chow, were tested in a spontaneous breast cancer model in 34 mice. Subgroups of 3–9 mice were assigned, in which the diet were implemented either with or without added rapamycin, an mTOR inhibitor and potential anti-cancer drug.

### Results

Blood glucose and insulin concentrations in mice ingesting the ketogenic diet (KD) were significantly lower, whereas beta hydroxybutyrate (BHB) levels were significantly higher, respectively, than in mice on the standard diet (SD). Growth of primary breast tumors and lung metastases were inhibited, and lifespans were longer in the KD mice compared to mice on the SD ($p < 0.005$). Rapamycin improved survival in both mouse diet groups, but when combined with the KD was more effective than when combined with the SD.

### Conclusions

The study provides proof of principle that a ketogenic diet a) results in serum insulin reduction and ketosis in a spontaneous breast cancer mouse model; b) can serve as a therapeutic anti-cancer agent; and c) can enhance the effects of rapamycin, an anti-cancer drug, permitting dose reduction for comparable effect. Further, the ketogenic diet in this model produces superior cancer control than standard mouse chow whether with or without added rapamycin.

which operates in the real estate sector. In addition, the study was supported in part by the CTSA Grant UL1TR002556 from the National Center for Advancing Translational Sciences (NCATS), a component of the National Institutes of Health (NIH). The funders had no role in study design, data collection and analysis, decision to publish, or preparation of the manuscript.

**Competing interests:** Our funder, ST Balchug, is a commercial source acting in the real estate sector. This does not alter our adherence to PLOS ONE policies on sharing data and materials.

## Introduction

Insulin inhibition by a ketogenic diet has been shown to slow cancer growth and prolong survival in animal models and has shown safety and feasibility in small pilot studies in humans [1–4]. We previously demonstrated in a pilot study of ten people with diverse, metastatic PET positive cancers that higher levels of ketosis correlated with stability vs. disease progression throughout the course of the 28 day trial [5], Further, the metabolic rationale for ketogenic diets and insulin inhibition in cancer control is highly plausible [2,6]. The full potential of ketosis in cancer therapy, however, may reside in its potential to synergize with anti-cancer drugs and other modalities of treatment. Increased overall synergies may permit lower drug doses, thereby reducing their toxicities and side effects. Accordingly, it may be possible that the overall improvement in therapy will result in extended survival with a better quality of life.

An understanding of ketogenic diets (KD) in cancer is limited at this point but it seems unlikely that KDs by themselves can control all the features of the oncogenic state. There is much interest, therefore, in the possibility of synergy with other drugs or other therapies. Hsieh, et al., for example, demonstrated that a squamous cell carcinoma that over expressed GLUT1 receptors showed attenuated growth when animals were on a KD [7]. There was, however, little regression of tumors. The addition of the cytotoxic agent cisplatin to mice on a KD led to regression to a greater extent than cisplatin alone. An interesting variation of this principle was recently demonstrated in mice bearing a Kras-Tp53-Pdx-Cre (KPC) mutation by coupling a ketogenic diet with a PI3K inhibitor [8]. The specific benefit in this latter case was shown to arise from ketogenic diet attenuation of hyperglycemia induced by the PI3K inhibitor. Control of the hyperglycemia resulted in reduced glucose-driven cancer growth and proliferation. Hyperglycemia is also a well-known side effect of rapamycin in humans. Rapamycin, an antifungal compound known also as sirolimus, and congeners such as temsirolimus have been proposed and studied as anti-cancer drugs in triple negative breast cancers [9], but their usefulness may be limited by hyperglycemic side effects.

Rapamycin, in further analogy with PI3K inhibitors, has the potential to be an anti-cancer drug via its inhibition of mTOR, a signaling molecule downstream of PI3K which promotes cell growth and inhibits apoptosis. It has not achieved much clinical use due to hyperglycemic effects in humans [10]. Rapamycin causes diabetes in mice [11], although only at very high doses and extended duration of treatment. Rapamycin was selected for the current animal study because it a) has the potential to be a successful anti-cancer drug when administered at doses known to be normo-glycemic in mice b) it may yet have utility in humans and c) has been an FDA approved drug since 1999 [12]. The principle of combining a ketogenic diet with other forms of anti-cancer chemotherapies for widely metastatic disease has been reported to have potential additive effects in mice [13,14], as well as in limited human studies [4,15–18]. As above, rapamycin is not expected to induce hyperglycemia in our mouse model and we wished to determine if a KD would exert its cytotoxic effects even in the absence of hyperglycemia.

## Material and methods

The IACUC of the Albert Einstein College of Medicine reviewed and approved of this research protocol; protocol number 20170408.

### Diets

Diets were purchased from Research Diets Inc. The ketogenic diet (KD) composition in calorie percent ratio of carbohydrate/fat/protein was 0.1/89.9/10.0. The standard diet's (SD) distribution was 80/10/10. Both diets contain the same quality and quantity of mineral and vitamins

and other necessary components. The fat content in both diets was from cocoa butter, with 59.7 gm, 32.9 gm and 3.0 gm of saturated (33.2 stearic and 25.4 palmitic), monounsaturated (32.6 oleic) and polyunsaturated fat (2.8% linoleic), respectively, per 100 gm of total fat, with trace amounts of other fats making up about 3% of the remainder. Omega 6 and omega -3 fatty acids contributed 2.8 gm and 0,1 gm, respectively. The KD and SD contain 6.71 and 3.85 calories/gm of energy, respectively. In general, a mouse needs 13.7 to 14.6 calories from their food [19].

## Cancer model and treatment

Four-week old, female FVB/N-Tg(MMTV-PyVT)634Mul/J mice were purchased from The Jackson Laboratories. These mice (100%) develop breast tumors spontaneously during their lifetime. The breast tumors can be seen as early as 5 weeks of age. At four months, 80–94% of these mice will have developed lung metastasis [20,21].

Mice were randomly divided into 6 groups of mice, 34 in total, 17 designated to overall SD groups, 17 to KD groups. All mice were were held at the Animal Institute for one week where they were all administered a SD. After an additional week, i.e. at approximately 2 weeks after arrival, or six weeks after birth, animals were returned to the investigator and were immediately assigned to the dietary groups, namely the SD, SD plus rapamycin at 0.4 mg/kg (SDr0.4), SD plus rapamycin at 4 mg/kg (SDr4), KD, KD plus rapamycin at 0.4 mg/kg (KDr0.4), and KD plus rapamycin at 4 mg/kg (KDr4). The number of mice in these groups were 9,3,5,9,4 and 4, respectively. (Variation in expected numbers was unintentional, but resulted from inadvertent deaths of several animals due to a novice animal husbander). At the third week of the special diets, rapamycin was given to mice by oral gavage with a 22-gauge feeding needle at a dose of 0.4 mg/kg or 4 mg/kg daily for 2 weeks. Mice were then maintained on both diets until euthanasia was required.

## Housing and husbandry

The Albert Einstein College of Medicine has an AAALAC accredited animal facility with clean barrier housing for mice. Animal caretakers check mice daily and food and water are provided ad libitum except as required in IACUC approved animal protocols. Routine environmental enrichment includes housing of groups of up to five mice per cage with provision of cotton fiber nestlets or small huts. Temperature and humidity is constantly monitored and kept within acceptable ranges (68–72 degrees F and 30–70%). Three veterinarians and 4 veterinary technicians provide oversight and veterinary care. IACUC provides animal welfare oversight. Our mice were housed and husbanded in the institution's barrier animal facility. This was not secondary to intrinsic immunocompromise, as in nude mice. It was based, rather, on a requirement specific to the Institutional Animal Care and Use Committee (IACUC) of our institution: that all cancer mice, particularly those receiving chemotherapy (which may compromise the immune system secondarily), must be housed in our institutional barrier to reduce the infection rate. As we employed a spontaneous cancer mouse model in which all mice developed breast cancer after 5 weeks of age and some received chemotherapy, the institutional IACUC required barrier housing. Special training of the first author (YZ) was provided in animal handling, anesthesia, tumor measurement, moribund determination, oral gavage, and cardiocentesis.

## Blood sampling and analysis

At designated time points, mice were bled from the tail vein with a 21 G injection needle puncture. The peripheral blood drops were used to measure glucose and Beta-hydroxybutyrate

(BHB) separately using Keto Mojo, a blood glucose and ketone monitoring system. Each glucose data point is a daily average of three measurements at 3 different time points 9 am, 1 pm, and 5 pm. The Keto Mojo assay has been validated in an independent study by Augusta University [23] of the University of Georgia health system and further confirmed against the Beckman Coulter AU480 Chemistry Analyzer at Biomarker Analytic Research Core of Albert Einstein College of Medicine.

The serum obtained after cardiocentesis at the time of euthanasia (see below) was used to measure insulin level with an ELISA based on chemiluminescence.

### Determination of moribund status and humane endpoints

Animals were examined daily (by YZ) for their overall condition and signs of moribund behavior, and weekly to measure tumor volume. Mice were determined to have reached moribund status when they could no longer reach their food and/or when the sum of tumor volume within a mouse exceeded 4 cm$^3$. Once mice reached this condition, they were euthanized within four hours which then constituted the duration of the experiment. All (n = 34) animals were euthanized; none were euthanized prior to reaching this point. Moribund mice were euthanized in accordance with IACUC recommendations as well as with ARRIVE guidelines for humane endpoints. All animal welfare considerations were taken, including minimization of suffering and distress, including special barrier housing, as described above. They were anesthetized with 2.5% isoflurane, and blood ($\geq$1 ml) was taken by cardiocentesis resulting in immediate death.

### Tumor size and survival measurement

Tumor size was measured weekly with calipers (by YZ). Volume was calculated from the longest (L) and shortest (S) dimensions according to $Volume\ (mm^3) = \frac{1}{2}L \times S^2$. In this spontaneous breast cancer model, each mouse develops multiple tumors. The combined tumor volume represented by the sum of all visible tumor volumes was used as a surrogate measure of the overall tumor growth rate. (This measure does not include additional growth due to metastases). Longevity was determined when a mouse attained a morbid state, characterized by the inability to reach food and water normally, or when the sum of its tumor volume exceeded 4 cm$^3$. Moribund mice were euthanized based on these IACUC approved criteria of our institution. The lung and other major organs were resected and weighed, and the lung-to-body weight ratio was calculated. The tissues were fixed with 10% formalin and prepared for later blinded pathological evaluation.

Comparisons between SD vs. KD groups with respect to tumor size, and lung metastasis weight were performed using non-parametric Mann Whitney U tests. Serum measurement of insulin, BHB, and glucose were compared between groups using unpaired Student t-tests, also used for body weight comparisons. Longevity between groups was compared using log-rank testing.

### Statistics

Independent variables are between subjects. We used Prism 8 graph and statistical software. The threshold for statistical significance was 0.05 (5% confidence level). The Log-rank (Mantel-Cox) test was used for survival data, and two-tailed t-test or F-test for tumor weight and insulin data because of the normality of distributions with appropriately tight standard deviations. These data were double checked using Mann Whitney U testing. No additional codes were used in the analysis; multiple comparisons were compared using ANOVA for body weight only. No data were transformed, outliers were not removed, and there were no missing or excluded data.

## Results

### KD and SD had similar effects on body weight of mice

Six-week old, female mice were randomly divided into 6 groups, totaling 34 mice as described in Cancer Model, above. Three groups were assigned to a standard diet (SD) and three to a ketogenic diet. Within each diet group, after 2 weeks, rapamycin was administered by oral gavage with 22 gauge feeding needle at a dose 0.4 mg/kg or 4 mg/kg daily for 2 weeks.

The mice in all SD groups and KD groups (with or without rapamycin) were given the same caloric energy from start point to the end (< 8 weeks). As shown in Fig 1A, the body weights of the mice in KD groups were not significantly different from those in SD groups at any time point during the study.

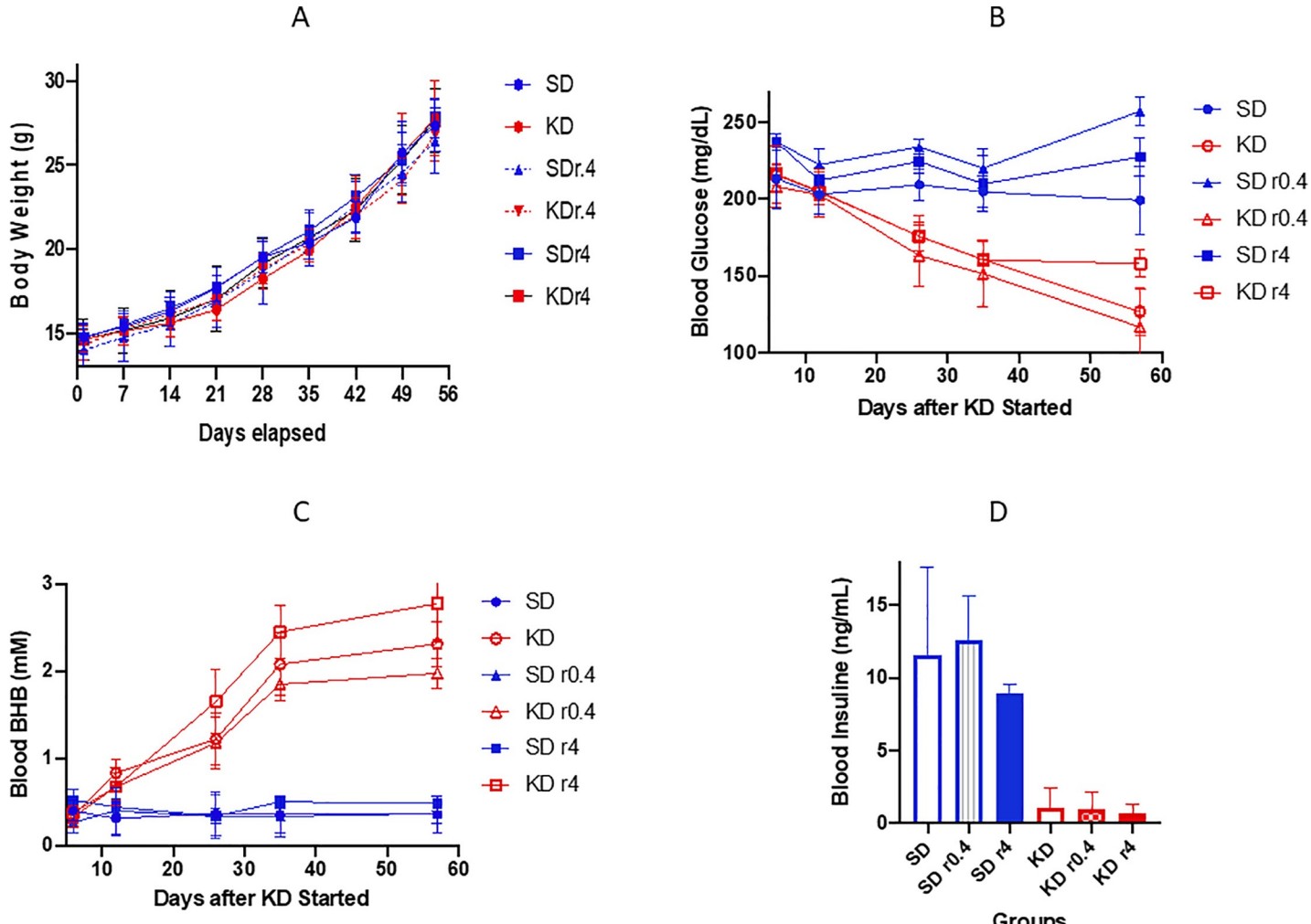

**Fig 1.** The body weight (A), blood glucose (B), beta hydroxybutyrate (C), and insulin (D) levels are displayed (A) Body weight increased equally without differences in all groups. (B) Each glucose data point is a daily average of 3–9 mice, and each mouse was measured 3 times in that particular day at 3 different time points (9 am, 1 pm, and 5 pm). (C) Each beta hydroxybutyrate data point is a daily average of 3–9 mice with single measurement per mouse. (D) The insulin levels were measured when the mice were moribund. The serum was used to measure insulin level with an ELISA method. The blue lines or bars represent the data from SD groups. The red lines or bars are the data from KD groups. r0.4 and r4 means rapamycin at the dose 0.4 mg/kg and 4 mg/kg for 2 weeks, respectively.

## KD reduced the blood glucose level in mice

Blood serum glucose concentrations were measured in all mice after one week of dietary intervention and at 4 additional intervals (Fig 1B). Glucose was lower in all KD groups (KD, KD r0.4, and KD r4)) than corresponding SD groups, and decreased further during 3 weeks of KD feeding. Blood glucose was significantly lower than that of mice in all SD groups at all time points. Rapamycin at a low dose did not have a clear effect on the glucose level. A higher dose of rapamycin (4 mg/kg) enhanced the glucose levels slightly in both SD r4 and KD r4. At day 57 the mildly elevated glucose ratios of KD r4/KD and SD r4/SD were 158/127 (p = 0.0042) and 227/199 (p = 0.0223), respectively (Fig 1B).

## KD increased the blood beta-hydroxybutyrate level in mice

After 3 weeks, mice in all KD groups showed at least a four-fold elevation of serum BHB concentrations compared with SD mice. These elevations all reached statistical significance (p <0.005). Rapamycin did not affect the BHB level (Fig 1C).

## KD reduced the blood insulin concentration in mice

At study termination (see Methods), we collected the blood from each mouse and measured insulin levels. As Fig 1D demonstrates, the insulin serum concentrations in all SD mice groups were 8 to 20-fold higher than the levels in the respective KD mice groups (p<0.0005). The paired comparison is shown in Table 1.

## KD inhibited tumor growth and prolonged mouse longevity

Mice were fed SD or KD from approximately 6 weeks of age until they reached a moribund condition. All tumor sizes were measured in each mouse weekly from day 1 to day 56, the time period during which no mice had yet reached moribundity. The combined tumor volumes (sum of all measurable tumor volumes in each mouse) of KD mice were smaller than that of SD mice (mean KD 506 mm$^3$ vs, mean SD 1262 mm$^3$, p <0.0001, 2- way ANOVA; Fig 2A). Rapamycin further enhanced the tumor growth inhibition: the mean combined tumor volume of KD r0.4 and KD r4 versus KD was 359 mm$^3$ vs. 506 mm$^3$ (p = 0.0049) and 195 vs. 506 (p < 0.0001), respectively.

As shown in Fig 2B, the median survival of KD mice increased to 78 days as compared to 65 days for SD mice, a 20% increase (p = 0.0002, log rank test). KD, when combined with rapamycin at dose 4 mg/kg further increased the median survival to 95 days when compared to KD diet alone (78 days, as above, p = 0.002); and vs. SD r4 (81 days, p = 0.0049). See **Fig 2**.

## KD reduced metastases in the lungs of mice

Moribund mice were euthanized and their major organs were resected as described in Methods. Only lungs were found to have metastatic tumors among all organ systems. Lungs were weighed before further pathological evaluation. The lung/body weight ratio of each mouse was

**Table 1. Blood insulin level and comparison.**

| Compared Groups | Mean Insulin Levels (ng/ml) | P Value |
| --- | --- | --- |
| SD vs KD | 11.55: 1.08 | 0.0001 |
| SD r0.4 vs KD r0.4 | 12.55: 0.96 | 0.0039 |
| SD r4 vs KD r4 | 8.90: 0.62 | <0.0001 |

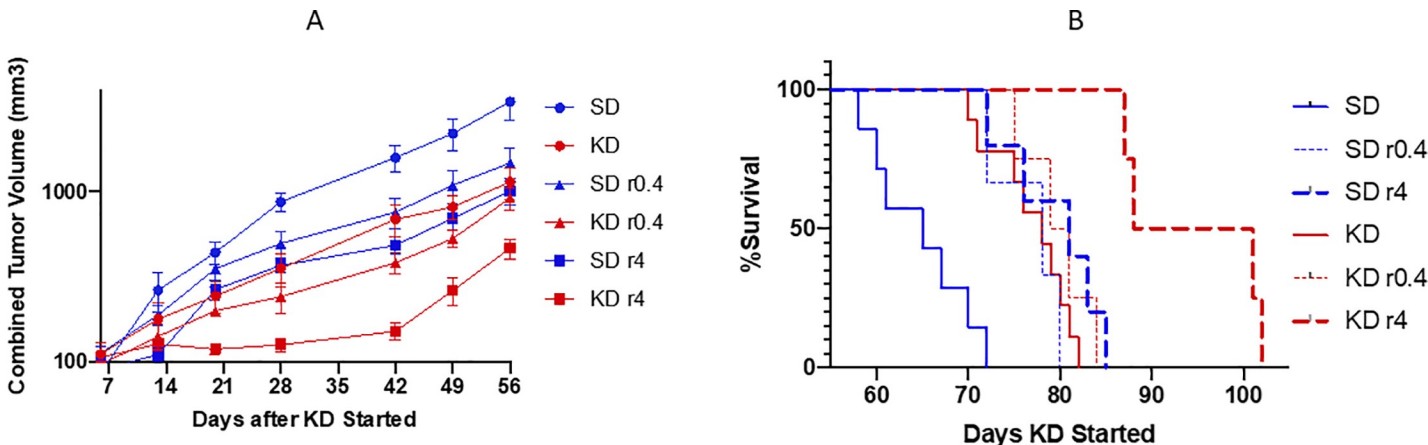

**Fig 2. Tumor size and survival.** Animal numbers at start of study diet: SD = 9, SD+r0.4 = 3; SD +r4 = 5; KD = 9; KD+r0.4 = 4; KD+r4 = 4; Tumor size (A) was measured once a week. Volume (mm$^3$) was calculated as 0.5 L x S$^2$ (L and S is the longest and shortest dimensions). The sum of all visible tumor volumes in each mouse was used as its tumor volume, and each point represents tumor volumes from 3–9 mice. Longevity (B) was determined to the time a mouse became moribund. The blue and red lines are the data from SD and KD groups, respectively. r0.4 and r4 represents rapamycin at the dose 0.4 mg/kg and 4 mg/kg for 2 weeks, respectively.

also calculated as it is positively related to the lung tumor number (n) and/or size [22]. The data is shown in **Fig 3A** and Table 2.

There is a significant difference of average lung/body weight ratios between SD and KD groups: 26.0 ± 2.2 mg/g vs. 15.3 ± 3.3 mg/g, p < 0.0001. SD/KD with rapamycin at the lower dose of 0.4 mg/kg trended toward further inhibition of lung metastasis (p = 0.0542), with lung/body weight ratio 23.1 ± 3.3 vs. 15.4 ± 4.5 mg/g. Rapamycin at a higher dose (4 mg/kg), when combined with KD showed more significant reduction of lung metastases: SD r4 vs. KD r4 was 16.4 ± 2.5: 10.2± 1.9 mg/g, p = 0.0045, and KD vs. KD r4 was 15.3 ± 3.3: 10.2 ± 1.9, p = 0.0163, respectively. See **Fig 3.** (above).

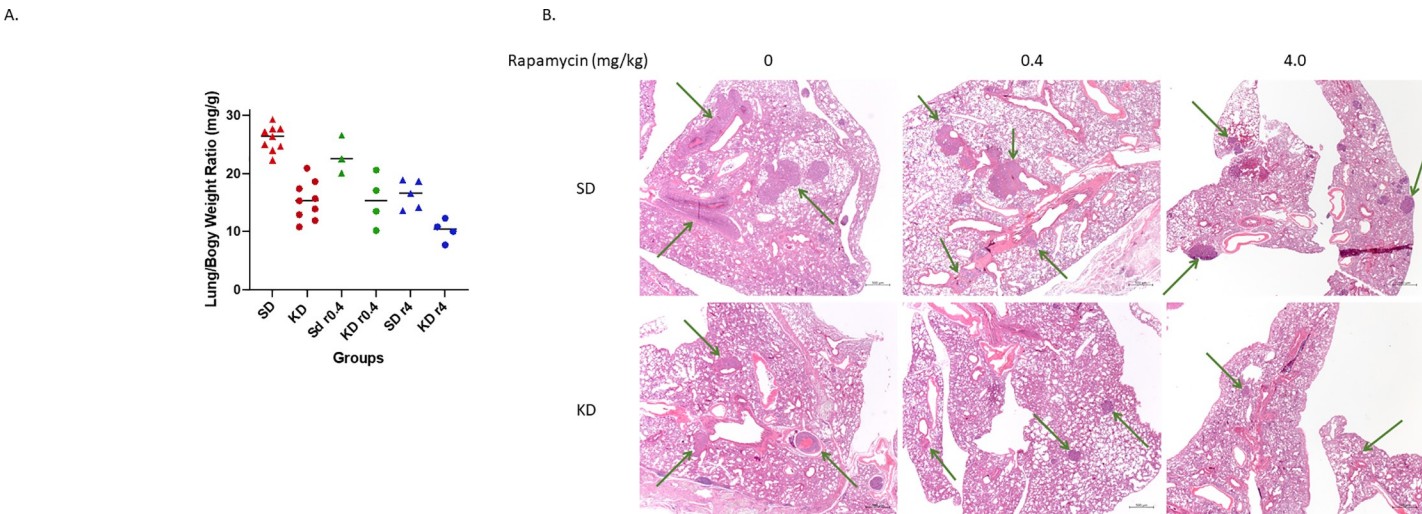

**Fig 3. The lung metastases.** Moribund mice were weighed, and their lungs were resected and weighed after taking ≥ 1 ml blood out from cardiac puncture. (A) The lung-to-body weight ratio from each group is presented. Round dots represent KD groups. Triangle dots represent SD groups. r0.4 and r4 represent rapamycin at doses of 0.4 mg/kg and 4 mg/kg separately. (B) The lung tissue pathology images are shown. The top row demonstrates lung tissue sections from SD, SDr0.4, and SDr4 groups. The bottom row shows lung tissue sections from KD, KDr0.4, and KDr4. The arrows point out tumor nodules. The magnification is 2.5 x for all pictures. The overall mass of tumors is reduced in the KD images.

Table 2. Lung/body weight ratios (mg/g).

| Comparison | Average Ratio | P Value |
|---|---|---|
| SD/KD | 26.0/15.3 | < 0.0001 |
| SD r0.4/KD r0.4 | 23.1/15.4 | 0.0542 |
| SD r4/KD r4 | 16.4/10.2 | 0.0045 |
| KD/KD r4 | 15.3/10.2 | 0.0163 |

The size and number of lung metastatic nodules were also measured in microscopic tumor slices from all mice. See S1 Table. These data provide general, but not statistically significant support for smaller overall metastases in all groups of the KD vs. SD animals. There is clearly limited statistical power due to small sample size; there are also large standard deviations. Both therapy groups have similar numbers of metastases. deviation.

## Discussion

We have tested the effect of two diets, one very low in carbohydrate ($\leq$ 0.1% of Calories); the other a standard diet (i.e. 80% calories from carbohydrate), on cancer growth and animal survival in a spontaneous breast cancer model in mice. Strict limitation of carbohydrate was effective at inhibiting serum insulin and glucose and inducing ketosis. While there were no 'cures', average overall tumor mass was reduced. The ketogenic group demonstrated prolonged survival as well. We also compared the effects of two different doses of rapamycin when added to the diets. Rapamycin delayed tumor growth in a dose-dependent manner and improved survival in both dietary groups. Greatest effects were seen when KD and rapamycin were combined.

In all groups, with or without rapamycin, the KD mice demonstrated longer survival, lower serum glucose and lower insulin concentrations than the corresponding SD mice. The total lung tumor mass in the KD animals was significantly and substantially smaller than in the corresponding SD mice. In view of significantly longer survival of the KD animals, both of these results are consistent with slower tumor growth in KD vs. SD.

Strict insulin inhibition can result in two principle effects, both of which have the potential to induce cancer cell programmed cell death and to reduce proliferation of cancer cells. First, reduced blood insulin concentrations at the cancer cell membrane results in less binding to the insulin receptor with resulting downstream inhibition of the PI3K-Akt-mTOR (PAM) signaling cascade [23], as well as the RAS-RAF-MEK-ERK pathway [24]. We therefore propose that reduced insulin concentration due to a ketogenic diet provides the potential to enhance programmed cell death by inhibiting the PAM cascade, and to reduce proliferation via both pathways [25,26]. A general caveat, of course, is that well-known, common mutations causing constitutive activation of PAM protein signals (e.g. PI3KCa) will resist programmed cell death and allow proliferation to continue [27]. Nonetheless those malignancies without these mutations can be therapeutically susceptible to the insulin inhibiting effects.

Second, hepatic ketogenesis due to insulin inhibition increases blood levels of the ketone bodies beta-hydroxybutyrate and acetoacetate, both of which have demonstrated histone deacetylase inhibitor effects at the cellular level. HDAC inhibitors are known to be capable of reducing cancer cell proliferation as well as enhancing programmed cell death [28–30].

Reduced proliferation was indeed observed as seen in the reduced extent of lung metastatic mass to total body mass ratios with KD in all groups. However, while necrosis in the KD groups was also detected in tumor specimens, meaningful differences with SD could not be identified definitively with our small sample size.

The potential of ketogenic diets to inhibit cancers has been suggested, mainly in animal models, for at least four decades [31,32]. Human data has been limited mostly to a few case reports or small clinical trials [5,33–36]. Additional studies have been reported more recently [37]. Meanwhile, in the past decade, interest has grown in insulin inhibition as a potential cancer therapeutic adjunct. Metformin, for example, has been applied toward this end with promising results in the neoadjuvant setting for breast cancer [38], but when applied therapeutically has demonstrated limited benefit [39]. Dietary carbohydrate restriction and metformin both reduce insulin secretion and glucose concentration, but the overall effect of carbohydrate restriction has additional effects beyond those of metformin. This is observed in its ability to induce formation of ketone bodies, known HDAC inhibitors. Ketone body formation speaks to the potential for translation to humans, as the 2 mM maximum extent of ketosis achieved in our mouse model is exceeded in people for whom 4–5 mM are quite achievable.

Nonetheless, one cannot assume that our mouse model results will translate to humans. Many mouse models, including our spontaneous breast tumor mouse with a B6 phenotype background, have been shown to have a degree of insulin resistance [40]. This resistance is reflected by a) persistent, high glucose concentrations in the SD group as well as b) delayed return of glucose concentrations to normal in the KD group (see persistent high glucose at 11 days in Fig 1, suggesting ongoing insulin resistance effect). Therefore, some of the relative survival disadvantage of the SD group may have resulted from a heightened insulin growth effect on the tumors. But it is worth noting, too, that the KD group might also have lived longer without prolonged high glucose levels. In any event, obesity with insulin resistance is commonplace among many patients with breast cancer, particularly in post-menopausal women [41]. The ultimate response to a KD in humans with breast cancer therefore demands controlled trials for those with normal as well as abnormal insulin sensitivity.

The results support the potential value of ketogenic diets in cancer therapy when coupled with existing agents, permitting additive or synergistic effects with toxic drugs. Successful translation, then, could result in reduction of drug doses while improving overall therapeutic effectiveness, thus extending patient survival while improving the quality of life during that period of greater longevity.

A ketogenic diet combined with existing drugs may provide a promising approach to increase the therapeutic effects of existing cancer therapies at lower levels of overall toxicity.

## Supporting information

**S1 Table. Size and number of metastases in lungs from microscopic slide.**
(DOCX)

**S1 File.**
(DOCX)

**S2 File.**
(DOCX)

**S3 File.**
(DOCX)

## Acknowledgments

The authors are grateful to Linda Jelicks and Wade Koba for helpful suggestions and consistent support and to Dr. Igor Koman for his contribution and support.

## Author Contributions

**Conceptualization:** Alexander Pearlman, Eugene J. Fine.

**Formal analysis:** Yiyu Zou, Susan Fineberg, Eugene J. Fine.

**Funding acquisition:** Eugene J. Fine.

**Investigation:** Yiyu Zou, Eugene J. Fine.

**Methodology:** Yiyu Zou.

**Project administration:** Eugene J. Fine.

**Resources:** Richard D. Feinman, Eugene J. Fine.

**Supervision:** Eugene J. Fine.

**Validation:** Yiyu Zou.

**Writing – original draft:** Yiyu Zou, Eugene J. Fine.

**Writing – review & editing:** Yiyu Zou, Alexander Pearlman, Richard D. Feinman, Eugene J. Fine.

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
