## [Decision Letter · Decision Letter 0]

2 Jun 2020

PONE-D-20-11420

The Effect of a Ketogenic Diet and Synergy with Rapamycin in a Mouse Model of Breast Cancer.

PLOS ONE

Dear Dr. Fine,

Thank you for submitting your manuscript to PLOS ONE. After careful consideration, we feel that it has merit but does not fully meet PLOS ONE’s publication criteria as it currently stands. Therefore, we invite you to submit a revised version of the manuscript that addresses the points raised during the review process.

Both reviewers have identified a number of issues which need to be addressed if the authors plan to submit a revised manuscript. The final comment of reviewer 2 in particular needs to be considered. As you can see the reviewer seeks clarification as to how the measurements could have been blind if only one individual othdid all of the measurements.

We look forward to receiving your revised manuscript.

Kind regards,

Salvatore V Pizzo

Academic Editor

PLOS ONE

Journal Requirements:

2. At this time, we request that you  please report additional details in your Methods section regarding animal care, as per our editorial guidelines:

(a) Please state the specific source of the mice used in the study (e.g. Jackson laboratories). In addition, please state the total number of mice used in the study

(b) Please provide details of animal welfare (e.g., shelter, food, water, environmental enrichment)

Thank you for your attention to these requests.

3. Please note that PLOS does not permit references to “data not shown.” Authors should provide the relevant data within the manuscript, the Supporting Information files, or in a public repository. If the data are not a core part of the research study being presented, we ask that authors remove any references to these data.

4. To comply with PLOS ONE submission guidelines, in your Methods section, please provide additional information regarding your statistical analyses. For more information on PLOS ONE's expectations for statistical reporting, please see https://journals.plos.org/plosone/s/submission-guidelines.#loc-statistical-reporting

'The authors are very grateful to ST Balchug who provided private philanthropic funding for these investigations. In addition, the study was supported in part by the CTSA Grant UL1TR002556 from the National Center for Advancing Translational Sciences (NCATS), a component of the National Institutes of Health (NIH)'

'The funding for this study and for all the authors was provided by a private philanthropic donor (mentioned in the Acknowledgments). Accordingly there was no national grant award number. The sponsor does not have a website for philanthropic giving. The sponsor made no effort to influence any aspect of the study design, data collection and analysis, decision to publish, or preparation of the manuscript.'

b. Additionally, because some of your funding information pertains to commercial funding, we ask you to provide an updated Competing Interests statement, declaring all sources of commercial funding.

In your Competing Interests statement, please confirm that your commercial funding does not alter your adherence to PLOS ONE Editorial policies and criteria by including the following statement: "This does not alter our adherence to PLOS ONE policies on sharing data and materials.” as detailed online in our guide for authors  http://journals.plos.org/plosone/s/competing-interests.  If this statement is not true and your adherence to PLOS policies on sharing data and materials is altered, please explain how.

c. Please include the updated Competing Interests Statement and Funding Statement in your cover letter. We will change the online submission form on your behalf.

6. We note that you have indicated that data from this study are available upon request. PLOS only allows data to be available upon request if there are legal or ethical restrictions on sharing data publicly. For information on unacceptable data access restrictions, please see http://journals.plos.org/plosone/s/data-availability#loc-unacceptable-data-access-restrictions.

7. Please include captions for your Supporting Information files at the end of your manuscript, and update any in-text citations to match accordingly. Please see our Supporting Information guidelines for more information: http://journals.plos.org/plosone/s/supporting-information

Reviewers' comments:

Reviewer's Responses to Questions

**Comments to the Author**

1. Is the manuscript technically sound, and do the data support the conclusions?

Reviewer #1: Yes

Reviewer #2: No

2. Has the statistical analysis been performed appropriately and rigorously? 

Reviewer #1: Yes

Reviewer #2: I Don't Know

3. Have the authors made all data underlying the findings in their manuscript fully available?

Reviewer #1: No

Reviewer #2: Yes

4. Is the manuscript presented in an intelligible fashion and written in standard English?

Reviewer #1: Yes

Reviewer #2: Yes

5. Review Comments to the Author

Reviewer #1: The paper "The effect of a ketogenic diet and synergy with rapamycin in a mouse model of breast

cancer" is an important contribution to the growing literature on ketogenic therapy in cancer.

The research question is well described and relevant given the fact that rapamycin has received a lot of attention as a

calorie restriction mimetic and anti-ageing drug.

The study has several strengths, among them the naturally occurence of the murine breast tumors instead of xenografted ones,

measurement of relevant outcome variables and good description of the methods and results.

The major limitation is the small sample size in some of the treatment groups (n=4), but the

treatments were chosen as to still provide sufficiantly clear outcome differences.

The most obvious limitation when reading the paper was that many references did not support the claims made.

I therefore recommend publication of this paper after some points are corrected by the authors. I think this will

further improve the scientific rigor and clarity of argumentation.

- Lines 37-39: The authors refer to insulin inhibition, suggesting therefby from the outset

that this is a major mechanism how ketogenic diets work. However, none of the four references cited

have measured insulin levels or discussed this mechanism in great detail. Ref. 1 has no authors listed , I suppose

it is a review by Allen et al. 2014, Redox Biol 2:963-970 ? Ref.4 did not show any benefit of the ketogenic diet

and therefore does not support the claim of the authors made in that sentence. Please cite more relevant studies showing

tumor growth inhibition potentially achieved via insulin decrease (e.g. Venkateswaran et al. 2007, JNCI 99:1793-1800)

or drop the statement about insulin here.

- Line 73: The authors refer to additive effects of a ketogenic diet with chemotherapy in mice, but cite Iyikesici et al (ref.12) which is

a human study, Gluschnaider et al. (ref.13) which did not combine a ketogenic diet with chemotherapy, and Schartz et al. (ref. 14) which is

a review not specifically on the ketogenic diet. Thus, none of these references support the claims made. Instead, I would recommend citing

the following studies:

Klement (2018), Complementary Medicine Research 25(2):102-113; Morscher et al. (2016), Oncotarget 7:17060–17073;

Kim et a. (2012), BJU International 110:1062-1069

PLease also delete "for widely metastatic disease" (line 72), this is not true for all studies

- Same line (73): Ref.4 and ref. 15 have shown no benefit of the ketogenic diet, hence do not support the claim made

- Line 103: Please define IACUC at first usage.

- Line 123: Instead of "etc." please name what has been tested

- Line 157: The t-test assumes normal distribution of the data. Have you tested if this is valid? I doubt that with such small sample size this

could be done. Instaed, I would suggest using the Mann Whitney U test here, too.

- Lines 172-173: Please read this sentence again, it appears there is a word missing and "mouse" should be "mice"

- Line 193 (caption to Fig. 1) The description of panel A (body weight) is missing, and (A) should be (B), (B) should be (C) and (C) should be (D).

- Line 210: 2-way ANOVA should also be mentioned in the Methods section

- Line 285: The abbreviation HDACi is never used after definition and can be dropped; in line 300 use "HDAC" instead of "histone deacetylase"

- Lines 293-294: "Human data has been sparse"  First RCTs have been conducted! Please cite Klement et al. 2020, Med Oncol 37:14 who

have summarized clincal studies with relevant outcomes

- Figure 1: axis label "Days elapsed" refers to which time point?

- Data availability statement was missing

Reviewer #2: The manuscript entitled “The Effect of a Ketogenic Diet and Synergy with Rapamycin in a Mouse Model of Breast Cancer.” PONE-D-20-11420 The authors attempt to demonstrate that ketogenic diet combined with rapamycin inhibits breast cancer growth. There are both major and minor problems with this paper that make a thorough review difficult.

Some minor problems that need to be addressed: Line 73 of introduction states, “effects in humans” and then lists references that are feasibility studies, and do not show effectiveness. Researchers must be very careful with language when discussing human trials. Line 82 list the protein/carb/fat content but does not discuss the kind of fat used. We have found that the lipid/PUFA content of the KD is very important for tumor growth inhibition. Please list the complete content of the diet. Line 183 a decimal is missing. Figure 1B, the initial blood glucose is very high in all the animals. Even after 11 days the glucose level is very high in KD groups. This is very surprising and counterintuitive. Sometimes blood glucose in mice spikes if they are scared and the time of the blood draw and you don’t act very quickly. Therefore, hemoglobin A1C might be a better indicator of true glucose control. Figure 1D insulin spelling on the axis and no indication of statistics performed. Figure 2A how many mice were in each group. It should be in the legend. The way that you measure tumor growth hides important data. Did you count the number of tumors arising on each mouse (tells us if initiation is affected); vs tumor volume of a single tumor (growth inhibition).

Some major problems to be addressed: line 108 indicates that there was only one investigator measure the mice. How could you keep that person blinded? Tumor size measurements has been shown to be subject to UNintentional biasing. It is essential that the investigator be blinded. Also, the general health of the mice needs to be investigated. Were they lethargic? Was there a change in their coat? Is there any other indication that the mice we made ill by the treatment? Lastly, the lung needs to be examined by a blinded pathologist to determine the number and stage of the nodules.

In general, there might be interesting information within the text but because the data is not adequately described it is impossible for me to adequately review it.

6. PLOS authors have the option to publish the peer review history of their article (what does this mean?). If published, this will include your full peer review and any attached files.

Reviewer #1: Yes: Rainer J. Klement

Reviewer #2: Yes: Melissa Fath

---

## [Author Response · Author response to Decision Letter 0]

19 Aug 2020

We believe we have observed such requirements.

At this time, we request that you please report additional details in your Methods section regarding animal care, as per our editorial guidelines:

(a) Please state the specific source of the mice used in the study (e.g. Jackson laboratories). In addition, please state the total number of mice used in the study

(b) Please provide details of animal welfare (e.g. shelter, food, water, environmental enrichment). 

Thank you for your attention to these requests.

Existing text was noted as Jax Lab (a subdivision of Jackson Laboratories which provide the animals), but changed to Jackson Laboratories. 

Total numbers of mice used now indicated, and animal welfare description expanded.

3. Please note that PLOS does not permit references to “data not shown.” Authors should provide the relevant data within the manuscript, the Supporting Information files, or in a public repository. If the data are not a core part of the research study being presented, we ask that authors remove any references to these data.

Done.

4. To comply with PLOS ONE submission guidelines, in your Methods section, please provide additional information regarding your statistical analyses. For more information on PLOS ONE's expectations for statistical reporting, please see https://journals.plos.org/plosone/s/submission-guidelines.#loc-statistical-reporting

'The authors are very grateful to ST Balchug who provided private philanthropic funding for these investigations. In addition, the study was supported in part by the CTSA Grant UL1TR002556 from the National Center for Advancing Translational Sciences (NCATS), a component of the National Institutes of Health (NIH)'

Removed from Acknowledgments

'The funding for this study and for all the authors was provided by a private philanthropic donor (mentioned in the Acknowledgments). Accordingly there was no national grant award number. The sponsor does not have a website for philanthropic giving. The sponsor made no effort to influence any aspect of the study design, data collection and analysis, decision to publish, or preparation of the manuscript.'

We’ve removed funding information from the Acknowledgments.

The Funding Statement will now reflect the philanthropic sponsor as well as the CTSA Grant UL1TR002556 from the National Center for Advancing Translational Sciences (NCATS), a component of the National Institutes of Health (NIH). 

b. Additionally, because some of your funding information pertains to commercial funding, we ask you to provide an updated Competing Interests statement, declaring all sources of commercial funding.

In your Competing Interests statement, please confirm that your commercial funding does not alter your adherence to PLOS ONE Editorial policies and criteria by including the following statement: "This does not alter our adherence to PLOS ONE policies on sharing data and materials.” as detailed online in our guide for authors http://journals.plos.org/plosone/s/competing-interests. If this statement is not true and your adherence to PLOS policies on sharing data and materials is altered, please explain how.

Above comments are incorrect. The philanthropic funding is not from a commercial source. The point above is therefore moot.

c. Please include the updated Competing Interests Statement and Funding Statement in your cover letter. We will change the online submission form on your behalf.

Not applicable.

6. We note that you have indicated that data from this study are available upon request. PLOS only allows data to be available upon request if there are legal or ethical restrictions on sharing data publicly. For information on unacceptable data access restrictions, please see http://journals.plos.org/plosone/s/data-availability#loc-unacceptable-data-access-restrictions.

We remove the statement that data are available on request. There are no relevant data required for the submitted paper, which is now complete as is.

The above statements are relevant to human studies, but not to an animal study.

7. Please include captions for your Supporting Information files at the end of your manuscript, and update any in-text citations to match accordingly. Please see our Supporting Information guidelines for more information: http://journals.plos.org/plosone/s/supporting-information

Reviewers' comments:

Reviewer's Responses to Questions

Comments to the Author

1. Is the manuscript technically sound, and do the data support the conclusions?

Reviewer #1: Yes

Reviewer #2: No

2. Has the statistical analysis been performed appropriately and rigorously? 

Reviewer #1: Yes

Reviewer #2: I Don't Know

3. Have the authors made all data underlying the findings in their manuscript fully available?

Reviewer #1: No

Reviewer #2: Yes

4. Is the manuscript presented in an intelligible fashion and written in standard English?

Reviewer #1: Yes

Reviewer #2: Yes

5. Review Comments to the Author

 

Reviewer #1: 

The major limitation is the small sample size in some of the treatment groups (n=4), but the treatments were chosen as to still provide sufficiEntly clear outcome differences.

As the reviewer says, the outcome is clear. The main point is that in studies like these we have control of the independent variables. There is clear separation of the different experimental groups

The most obvious limitation when reading the paper was that many references did not support the claims made. I therefore recommend publication of this paper after some points are corrected by the authors. I think this will further improve the scientific rigor and clarity of argumentation.

- Lines 37-39: The authors refer to insulin inhibition, suggesting thereby from the outset that this is a major mechanism how ketogenic diets work. However, none of the four references cited have measured insulin levels or discussed this mechanism in great detail. Ref. 1 has no authors listed , I suppose it is a review by Allen et al. 2014, Redox Biol 2:963-970 ? 

- Ref.4 did not show any benefit of the ketogenic diet and therefore does not support the claim of the authors made in that sentence. 

- 

Please cite more relevant studies showing tumor growth inhibition potentially achieved via insulin decrease (e.g. Venkateswaran et al. 2007, JNCI 99:1793-1800)

or drop the statement about insulin here.

We have changed the initial references. We are grateful to the reviewer for pointing us to Venkateswaran. References 1-4 are now: (Note: the text now refers to 1-7 since references 5-7 also refer to insulin involvement):

NEW REFERENCES

1. Venkateswaran, V. Haddad, A. Fleshner, NF, et al. Association of Diet-Induced Hyperinsulinemia With Accelerated Growth of Prostate Cancer (LNCaP) Xenografts, JNCI: Journal of the National Cancer Institute, 99, (23) 1793–1800, https://doi.org/10.1093/jnci/djm231

2. Klement and Kämmerer Is there a role for carbohydrate restriction in the treatment and prevention of cancer? Nutrition & Metabolism 2011, 8:75 http://www.nutritionandmetabolism.com/content/8/1/75

3. Goodwin PJ. Obesity, insulin resistance and breast cancer outcomes. Breast. 2015;24 Suppl 2:S56-9. Epub 2015/08/19. doi: 10.1016/j.breast.2015.07.014. PubMed PMID: 26283600. **FORMER REF 35**

6. Fine EJ, Feinman RD: Insulin, Carbohydrate Restriction, Metabolic Syndrome and Cancer. Expert Rev Endocrinol Metab 2014, 9(6).

- Line 73: The authors refer to additive effects of a ketogenic diet with chemotherapy in mice, but cite Iyikesici et al (ref.12) which is

a human study, Gluschnaider et al. (ref.13) which did not combine a ketogenic diet with chemotherapy, and Schartz et al. (ref. 14) which is

a review not specifically on the ketogenic diet. Thus, none of these references support the claims made. Instead, I would recommend citing

the following studies:

Klement (2018), Complementary Medicine Research 25(2):102-113; Morscher et al. (2016), Oncotarget 7:17060–17073;

Kim et a. (2012), BJU International 110:1062-1069

These are now references 12 and 13 and we refer only to these 2 and have moved reference 12 to reference 15 and we now include the references as (15-17)

PLease also delete "for widely metastatic disease" (line 72), this is not true for all studies

We are pointing out references which report on use of a ketogenic diet in widely metastatic disease, even if only some of their referenced papers report on these. The relative sparsity of data for KD’s in metastatic disease is the point.

- Same line (73): Ref.4 and ref. 15 have shown no benefit of the ketogenic diet, hence do not support the claim made

Reference 4 has been replaced with

1. Klement RJ. Beneficial effects of ketogenic diets for cancer patients: a realist review with focus on evidence and confirmation. Med Oncol. 2017;34(8):132. Epub 2017/06/28. doi: 10.1007/s12032-017-0991-5. PubMed PMID: 28653283.

Fine EJ, Feinman RD: Insulin, Carbohydrate Restriction, Metabolic Syndrome and Cancer. Expert Rev Endocrinol Metab 2014, 9(6)

- Line 103: Please define IACUC at first usage.

- 

 “IACUC (Institutional Animal Care and Use Committee)” has been inserted.

- Line 123: Instead of "etc." please name what has been tested

We have changed the text to read:

“was used to measure insulin level with an ELISA based on chemiluminescence.” 

We’ve eliminated text describing lab data not relevant to the study (BUN, bicarbonate, creatinine)

- Line 157: The t-test assumes normal distribution of the data. Have you tested if this is valid? I doubt that with such small sample size this could be done. Instaed, I would suggest using the Mann Whitney U test here, too.

We have added the Mann Whitney U test which confirms the results.

- Lines 172-173: Please read this sentence again, it appears there is a word missing and "mouse" should be “mice"

Changed to 

“As shown in Figure 1b., serum glucose concentrations decreased slightly in all mice after one week. A further decrease was observed in all KD groups (KD, KD r0.4, and KD r4), decreasing further for the remainder of period of KD feeding. “

- Line 193 (caption to Fig. 1) The description of panel A (body weight) is missing, and (A) should be (B), (B) should be (C) and (C) should be (D).

- Corrected, as below:

Fig 1. Comparison of KD and SD groups on (A) body weight,(B), blood glucose and (C) beta hydroxybutyrate and (D) insulin. Each glucose data point is a daily average of 3-9 mice, and each mouse was measured at 3 different time points (9 am, 1 pm, and 5 pm) each day. Each beta hydroxybutyrate data point is a daily average of 3-9 mice with single measurement per mouse. The insulin levels were measured with an ELISA method when the mice were moribund. The blue lines or bars represent the data from SD groups. The red lines or bars are the data from KD groups. r0.4 and r4 means rapamycin at the dose 0.4 mg/kg and 4 mg/kg for 2 weeks, respectively.

- Line 210: 2-way ANOVA should also be mentioned in the Methods section

Done.

Line 145.

The combined tumor volume represented by the sum of all visible tumor volumes was used as a surrogate measure of the overall growth rate of the primary tumors and was analyzed with 2-way ANOVA (fig. 2A) . (This measure does not include additional growth due to metastases) 

Also added to Legend which now reads:

Figure 2. Tumor size and survival. Tumor size (A) was measured once a week. Volume (mm3) was calculated as 0.5 L x S2 (L and S is the longest and shortest dimensions). The sum of all visible tumor volumes in each mouse was used as its tumor volume, and each point represents tumor volumes from 3-9 mice. mean KD 506 mm3 vs, mean SD 1262 mm3, p <0.0001, 2 way ANOVA

- Line 285: The abbreviation HDACi is never used after definition and can be dropped; in line 300 use "HDAC" instead of "histone deacetylase”

Changed.

- Lines 293-294: "Human data has been sparse"  First RCTs have been conducted! Please cite Klement et al. 2020, Med Oncol 37:14 who

have summarized clinlcal studies with relevant outcomes. 

New Reference 31. Old reference 31 now reference 35. We agree that we were understating the case, considering the accumulating evidence [31] and references therein; although the field must be considered in its early stages.

- Figure 1: axis label "Days elapsed" refers to which time point?

Label changed to DAYS AFTER KD STARTED

- Data availability statement was missing

“Data available from the authors upon request.” Has been added to end of MS.

 

Reviewer #2: 

The manuscript entitled “The Effect of a Ketogenic Diet and Synergy with Rapamycin in a Mouse Model of Breast Cancer.” PONE-D-20-11420 The authors attempt to demonstrate that ketogenic diet combined with rapamycin inhibits breast cancer growth. There are both major and minor problems with this paper that make a thorough review difficult.

Some minor problems that need to be addressed: Line 73 of introduction states, “effects in humans” and then lists references that are feasibility studies, and do not show effectiveness. Researchers must be very careful with language when discussing human trials.

As described in answer to Reviewer #1 we have changed the references.

Line 82 lists the protein/carb/fat content but does not discuss the kind of fat used. We have found that the lipid/PUFA content of the KD is very important for tumor growth inhibition. Please list the complete content of the diet.

The diet composition is now listed. Both diets contained fat derived from cocoa butter, obviously with the same ratio of saturated, monounsaturated and polyunsaturated fats; of course, the total amount of fat was very different. It is certainly plausible that other ratios of PUFA to saturated fats would have effects beyond those we’ve shown. It would be interesting and important to understand those effects, but that is a different study than we sought to accomplish.

 Line 183 a decimal is missing. 

Fixed.

Figure 1B, the initial blood glucose is very high in all the animals. Even after 11 days the glucose level is very high in KD groups. This is very surprising and counterintuitive. Sometimes blood glucose in mice spikes if they are scared and the time of the blood draw and you don’t act very quickly. Therefore, hemoglobin A1C might be a better indicator of true glucose control.

This is a good point and interesting observation. This is not a common breed of mouse and the high glucose may be related to the physiology but, at this point, it is a reproducible measurement. It has also been our observation in other mouse and rat experiments that high and irregular glucose concentrations can be observed. Despite the high glucose value, it was interesting, and we believe important, that insulin remained low in the ketogenic diet group, and not in the standard diet group. It appears that in at least some rodent breeds glucose levels are not regulated nearly as tightly as in humans. We did not measure glucagon, and perhaps that would have some bearing.

Figure 1D insulin spelling on the axis

Fixed. 

and no indication of statistics performed. Figure 2A how many mice were in each group. It should be in the legend.

Done.

 The way that you measure tumor growth hides important data. Did you count the number of tumors arising on each mouse (tells us if initiation is affected); vs tumor volume of a single tumor (growth inhibition).

**: “Because there are many tumors of different sizes, an ideal presentation is not obvious. It would surely be interesting to understand initiation better, but this was not the point of the study.

Some major problems to be addressed: line 108 indicates that there was only one investigator measure the mice. How could you keep that person blinded? Tumor size measurements has been shown to be subject to UNintentional biasing. It is essential that the investigator be blinded.

Tumor size was measured with calipers. We do not believe bias was involved but we must acknowledge that it’s possible to harbor an inadvertent bias, ie to ‘wish’ for a particular result. So we thank the reviewer for pointing out this possibility.

However, even if bias of any kind was involved at the time of caliper measurements, lung weight (vastly tumor) and body were weighed at post-mortem. There is no personal judgement involved in weight measurements. If bias was involved at the point of animal euthanasia (i.e. if tumors were measured as too large in order to inadvertently delay the euthanasia) then the weight measurement of metastases to the lungs should not have turned out lower than in the standard diet group.

The above said, we appreciate the insight. Whereas we did not even consider the idea of blinding the caliper measurements in the present study, we will certainly do so in future studies. I have double checked with the IACUC who have informed me that this is indeed the standard.

 Also, the general health of the mice needs to be investigated. Were they lethargic? Was there a change in their coat? Is there any other indication that the mice we made ill by the treatment? Lastly, the lung needs to be examined by a blinded pathologist to determine the number and stage of the nodules.

The general health of the mice was indeed evaluated daily, as in the text. In particular, the ability of the mice to reach their food or water (or lethargy, i.e. the inability to reach the food or water) was assessed regularly, as indicated. The coat was evaluated weekly, but no changes were observed. Since inability to reach food was the justification for euthanasia, as determined by our IACUC, we did not indicate animal coat quality in the text; a clear coat did not change our decision. The animals were dragging due to a huge burden of primary and especially metastatic disease, as their lung to total body weight ratios in our table indicate. The coat quality, which may otherwise be important, in this instance had no role in determining the endpoint of this study.

The lungs were indeed evaluated by a blinded pathologist (co-author Dr. Fineberg), as indicated in the text. But the most fundamental data, from the perspective of tumor metastatic mass, was the measurement of lung weight. A pathologic figure was provided, but only as representative of the lung data, as explicitly stated in the legend. Actual counting of mets and sizes of mets by the pathologist were statistically far less meaningful than overall metastatic lung mass and weight. The overall tumor mass provides the integral of tumor mass from what would have to be innumerable slides. The measure of total metastatic mass to determine overall metastatic information was approved by our pathology co-author who read and reviewed the manuscript, as indicated in our submission.

In general, there might be interesting information within the text but because the data is not adequately described it is impossible for me to adequately review it.

We respectfully disagree. We’ve added the additional information requested, but the data are qualitatively as well as quantitatively different between groups and highly consistent across experimental measurements. The experimental design determines how long the mice survived and the difference in metabolic parameters is clear. This is the point of the experiment. Representative tissue from pathologic sections are shown in Figure 4, but, as in the comment above, these kinds of data were not considered adequate for statistical evaluation of a process in a mouse model of breast cancer which by design metastasizes specifically to the lungs.

---

## [Decision Letter · Decision Letter 1]

4 Sep 2020

PONE-D-20-11420R1

The Effect of a Ketogenic Diet and Synergy with Rapamycin in a Mouse Model of Breast Cancer.

PLOS ONE

Dear Dr. Fine,

Thank you for submitting your manuscript to PLOS ONE. After careful consideration, we feel that it has merit but does not fully meet PLOS ONE’s publication criteria as it currently stands. Therefore, we invite you to submit a revised version of the manuscript that addresses the points raised during the review process.

Both reviewers found a number of minor issues which should be considered before the manuscript can be accepted for publication.

We look forward to receiving your revised manuscript.

Kind regards,

Salvatore V Pizzo

Academic Editor

PLOS ONE

Reviewers' comments:

Reviewer's Responses to Questions

**Comments to the Author**

1. If the authors have adequately addressed your comments raised in a previous round of review and you feel that this manuscript is now acceptable for publication, you may indicate that here to bypass the “Comments to the Author” section, enter your conflict of interest statement in the “Confidential to Editor” section, and submit your "Accept" recommendation.

Reviewer #1: (No Response)

Reviewer #2: (No Response)

2. Is the manuscript technically sound, and do the data support the conclusions?

Reviewer #1: Yes

Reviewer #2: Yes

3. Has the statistical analysis been performed appropriately and rigorously? 

Reviewer #1: Yes

Reviewer #2: Yes

4. Have the authors made all data underlying the findings in their manuscript fully available?

Reviewer #1: No

Reviewer #2: Yes

5. Is the manuscript presented in an intelligible fashion and written in standard English?

Reviewer #1: Yes

Reviewer #2: No

6. Review Comments to the Author

Reviewer #1: The authors have addressed all my previous comments satisfactorily except for one. In my previous comments I have stated:

Please cite Klement et al. 2020, Med Oncol 37:14 who

have summarized clinlcal studies with relevant outcomes.

However, the new reference 37 is not the correct one and has no relevance for the sentence in which it is used (line 323: "Additional trials have been reported more recently [37]"). Please change this reference.

Reviewer #2: (No Response)

7. PLOS authors have the option to publish the peer review history of their article (what does this mean?). If published, this will include your full peer review and any attached files.

Reviewer #1: **Yes: **Rainer J. Klement

Reviewer #2: No

---

## [Author Response · Author response to Decision Letter 1]

1 Oct 2020

The manuscript entitled “The Effect of a Ketogenic Diet and Synergy with Rapamycin in a Mouse Model of Breast Cancer.” PONE-D-20-11420 The authors attempt to demonstrate that ketogenic diet combined with rapamycin inhibits breast cancer growth. There are both major and minor problems with this paper that make a thorough review difficult. 

Some minor problems that need to be addressed: Line 73 of introduction states, “effects in humans” and then lists references that are feasibility studies, and do not show effectiveness. Researchers must be very careful with language when discussing human trials.

As described in answer to Reviewer #1 we have changed the references. 

I do not think you have adequately addressed my concerns. The first sentence and line 74 still imply effectiveness of KD in humans. None of the references 1-4 or 15-17 support effectiveness in human studies. There are numerous small studies that show feasibility of KD only. Reference 6 (probably changed for reference 5?) is a small pilot study that concludes nothing about prolonged survival. In addition, all references need to be double checked for accuracy. 

Reference 5 and reference 6 do indeed need to be swapped. We have done this.

We owe Dr. Klement an apology for improperly describing data that he illustrated very well in his review (ref. 37), now properly included. This review as well as some of the other articles now cited were not available on-line in our institution’s library at the time we wrote our response (the library as well as the institution were having IT difficulties, partly due to COVID19 and staffing) but these references have since become available. We were in the unfortunate position of quoting articles that Dr. Klement recommended without the opportunity to read them properly. 

In this comment I agree with the reviewer in many respects. We didn’t actually use the term effectiveness. The terms “effects” and “efficacy” are employed in several sentences, but not in reference to the articles Dr. Klement describes. The first sentence and line 74 both use the word “effects”, but in the context of animal studies, not humans. There is also a distinction between efficacy from effectiveness when used in a medical context. Effectiveness is an overall measure used when patients take drugs clinically, including an overall reduction of benefit when patients stop taking drugs due to side effects, for example. The term efficacy is limited to controlled trials, where behavior of the subjects is controlled, by definition, and where efficacy is measured among those who successfully take the medication. 

In any event our study’s goal was to reduce side effects so that an increase in the overall effectiveness of combined therapies might be realized, perhaps eventually in humans. So, perhaps ironically, we should actually have used the term effectiveness, not efficacy, with reference to our hoped for future objective in patients. We have swapped in that term in the relevant sentences of the Introduction as well as in the Discussion.

Given these distinctions, while we agree with the reviewer that phase 1 trials are designed to measure feasibility and safety and, are not designed to measure efficacy or effectiveness, it is not at all unusual for such measures to be reported anyway. For illustrative purposes, in refs 15-17, itemized by Dr. Klement, the trials were designed to measure safety and feasibility, but various outcome effects (overall survival, for example) were indeed measured and reported. 

I hope the above clarifies what we meant to say.

 Line 82 list the protein/carb/fat content but does not discuss the kind of fat used. We have found that the lipid/PUFA content of the KD is very important for tumor growth inhibition. Please list the complete content of the diet. 

The diet composition is now listed. Both diets contained fat derived from cocoa butter, obviously with the same ratio of saturated, monounsaturated and polyunsaturated fats; of course, the total amount of fat was very different. It is certainly plausible that other ratios of PUFA to saturated fats would have effects beyond those we’ve shown. It would be interesting and important to understand those effects, but that is a different study than we sought to accomplish.

Thank you for the though listing of fat/PUFA content. This is important data for other researchers wishing to build upon when answering the specific types of nutrients to include in a diet. 

Line 183 a decimal is missing. 

I did find a missing decimal on line 219 in the revised version and corrected this. Thank you. (I have line 183 in front of me and see no missing decimals. The number on line 183 is a statistical significance value of 0.05. This could be an issue of changing line numbers during editing.) 

Figure 1B, the initial blood glucose is very high in all the animals. Even after 11 days the glucose level is very high in KD groups. This is very surprising and counterintuitive. Sometimes blood glucose in mice spikes if they are scared and the time of the blood draw and you don’t act very quickly. Therefore, hemoglobin A1C might be a better indicator of true glucose control. 

This is a good point and interesting observation. This is not a common breed of mouse and the high glucose may be related to the physiology but, at this point, it is a reproducible measurement. It has also been our observation in other mouse and rat experiments that high and irregular glucose concentrations can be observed. Despite the high glucose value, it was interesting, and we believe important, that insulin remained low in the ketogenic diet group, and not in the standard diet group. It appears that in at least some rodent breeds glucose levels are not regulated nearly as tightly as in humans. We did not measure glucagon, and perhaps that would have some bearing.

Glucose and insulin level data if very important especially because this study might be translated into humans. It is important to discuss the limitations of model system. The original strain of mice in which this model system was derived, B6 have very high fasting and non-fasting glucose. They also have abnormal insulin responses. These mice might resemble diabetic or glucose intolerant humans more so than healthy humans. A good reference for this discussion is: Glucose Homeostasis and Tissue Transcript Content of Insulin Signaling Intermediates in Four Inbred Strains of Mice: C57BL/6, C57BLKS/6, DBA/2, and 129X1 Endocrinology, Volume 145, Issue 7, 1 July 2004. In Figure 1 the KD returns the glucose level down to what would be in wildtype strains or healthy humans not on a KD. For clarity please list the strain of mouse used in the abstract; add comparisons of glucose and insulin levels of wild type mice in the results; and the limitations to translation of the model system to humans in the discussion. Another important point for the discussion is that the diet in this study is very low in carbohydrates and does not resemble most human KDs. 

We thank the reviewer for pointing out our omission. We have surely added that effects seen in mouse models cannot be assumed to apply to humans. However, if anything, a higher glucose level in a mouse vs. a human (on a KD) would argue for a better effect in the human than the mouse. Humans also can reach higher levels of ketosis than mice, even though they consume less restrictive ketogenic diets, so the effects in mice again may argue for an even better response in humans. Nonetheless, we agree that we cannot assume translation to the human without further study and we have indicated this.

We have observed wide variations in blood glucose responses in mice in previous studies of our own. We have chosen to not discuss these aspects of mouse physiologic behavior in detail. This is a complex subject and has no direct bearing on the primary results showing prolonged survival. We’ve pointed out that rapamycin itself causes hyperglycemia in humans, although not in mice in doses such as those administered. The observation that hyperglycemia without hyperinsulinemia was seen initially on the ketogenic diet in this mouse model and then reverted to normal glucose concentrations during the ketogenic diet is surely an unusual finding. The overall pattern however shows distinct differences between a continued KD vs. a continued SD, where in the latter neither the high glucose levels (nor the insulin levels) correct over time. The suggestion that an initial anxiety reaction (provoking epinephrine secretion with attendant hyperglycemia, for example) is quite plausible, but we did not obtain epinephrine levels. 

Further discussion on the potential hormonal interactions would indeed be interesting but would nonetheless be largely speculative without additional data. At this point the speculation involved would be fodder for further criticism but would contribute little to our primary observations or the reasons for our study. This kind of discussion would be more appropriate to a comprehensive review article of hyperglycemia in mice and the variations seen in different mouse breeds.

In summary, the desired and expected physiologic effect of ketosis measured in our mouse model was, despite the extreme level of carbohydrate restriction, if anything, less pronounced than an expected effect in humans. We elicited 2 mM ketone body concentrations in mouse serum, whereas a lesser extent of CHO restriction normally achievable by humans can produce 4-5 mM or more. These differences between mouse and human models have no obvious bearing on our hypothesis or on our primary observations. Similar to the complex issue of glycemic responses in diverse mouse species above, this discussion would be more appropriate to a review article. 

Figure 1D insulin spelling on the axis and no indication of statistics performed. Figure 2A how many mice were in each group. 

 It should be in the legend. 

The number of mice in each group is not in the figure legend. The abstract should read 3-9 mice per group on line 20. Line 194 still incorrectly says 6 groups of 34 mice each. It is upsetting to this reviewer that the problem was not adequately addressed the first time. It looks like only 7 mice are in the survival graph in the SD. 

We apologize for not recognizing these errors, presumably resulting from the mental blur of reviewing the same text too many times. We’ve corrected the numbers in the paper as well as added the recommended numbers to the abstract. The numbers of mice in the SD at start are 9, as indicated.

The way that you measure tumor growth hides important data. Did you count the number of tumors arising on each mouse (tells us if initiation is affected); vs tumor volume of a single tumor (growth inhibition). 

**: “Because there are many tumors of different sizes, an ideal presentation is not obvious. It would surely be interesting to understand initiation better, but this was not the point of the study.

Fine, but if you have the lungs you could report the number of nodules. 

Thank you for your comment. We have included data from a pathologic slide that was evaluated. It is in our supplemental data. 

Tumor initiation is indeed interesting, and it is represented in our model by the spontaneous tumor initiation in breast tissue. Metastases do not reflect initiation as they arise due to additional mutations which permit breaking off from the primary tumor, lodging and growing in distant tissues, etc. The lung metastases were innumerable and these numbers were not reported by the pathologist. Larger nodule sizes were reported in the supplemental data, but these data were limited and differences between groups were not statistically significant, as indicated in the text. As you can see from the representative slide, many were microscopic. The bulk of the paraffin block was not used for staining. It was saved for RNA sequencing analysis and proteomics, both sent to other laboratories. These analyses were delayed by COVID and could not be included in this paper. At this point we don’t know if these samples will ever be analyzed or even recovered.

Some major problems to be addressed: line 108 indicates that there was only one investigator measure the mice. How could you keep that person blinded? Tumor size measurements has been shown to be subject to UNintentional biasing. It is essential that the investigator be blinded. Also, the general health of the mice needs to be investigated. 

Tumor size was measured with calipers. We do not believe bias was involved but we must acknowledge that it’s possible to harbor an inadvertent bias, ie to ‘wish’ for a particular result. So we thank the reviewer for pointing out this possibility. However, even if bias of any kind was involved at the time of caliper measurements, lung weight (vastly tumor) and body were weighed at post-mortem. There is no personal judgement involved in weight measurements. If bias was involved at the point of animal euthanasia (i.e. if tumors were measured as too large in order to inadvertently delay the euthanasia) then the weight measurement of metastases to the lungs should not have turned out lower than in the standard diet group. The above said, we appreciate the insight. Whereas we did not even consider the idea of blinding the caliper measurements in the present study, we will certainly do so in future studies. I have double checked with the IACUC who have informed me that this is indeed the standard.

Thank you for your considered response. I agree that the large difference in the tumor volume was unlikely due to unintentional bias. I also think that lung weight is valid measurement to report. The line about an individual measurer should be left in the manuscript so the limitations of the study are clearly understood. 

Thank you for your remark as well. We have indicated the individual measurer.

Were they lethargic? Was there a change in their coat? Is there any other indication that the mice we made ill by the treatment?

The general health of the mice was indeed evaluated daily, as in the text. In particular, the ability of the mice to reach their food or water (or lethargy, i.e. the inability to reach the food or water) was assessed regularly, as indicated. The coat was evaluated weekly, but no changes were observed. Since inability to reach food was the justification for euthanasia, as determined by our IACUC, we did not indicate animal coat quality in the text; a clear coat did not change our decision. The animals were dragging due to a huge burden of primary and especially metastatic disease, as their lung to total body weight ratios in our table indicate. The coat quality, which may otherwise be important, in this instance had no role in determining the endpoint of this study.

I understand that the condition of the mice regarding tumor burden was evaluated but what about diet and rapamycin. Did you see a change in the mice before the tumor endpoint that might indicate that they did not tolerate the diet or drug?

We’ve indicated all the changes that were observed. None of the already described changes could be attributed to rapamycin, and none suggested intolerance to either diet or to the drug. As mentioned, there were changes neither in the coat quality nor were there behavioral changes other than those already mentioned. We saw no intolerance to rapamycin or diet and saw no reason to raise these issues, particularly while mice have long been studied with ketogenic diets as well as with rapamycin at moderate doses without such intolerance reported. (Ketogenic diets administered by Tisdale et al, as in our reference go back to the 1980’s. Rapamycin has been used in mice studies for years, and just recently reported in Nature, (our ref. 8) without such effects noted.)

Lastly, the lung needs to be examined by a blinded pathologist to determine the number and stage of the nodules. 

The lungs were indeed evaluated by a blinded pathologist (co-author Dr. Fineberg), as indicated in the text. But the most fundamental data, from the perspective of tumor metastatic mass, was the measurement of lung weight. A pathologic figure was provided, but only as representative of the lung data, as explicitly stated in the legend. Actual counting of mets and sizes of mets by the pathologist were statistically far less meaningful than overall metastatic lung mass and weight. The overall tumor mass provides the integral of tumor mass from what would have to be innumerable slides. The measure of total metastatic mass to determine overall metastatic information was approved by our pathology co-author who read and reviewed the manuscript, as indicated in our submission. 

Line 172 should say blinded pathologist. For figure 4, the number of mets in the lung is interesting and important and should be included especially considering the experimental limitations on the other data. If the data is good enough to be included as a qualitative figure than it should be quantified. 

The number of overall mets was innumerable but was quantified per high power field by the pathologist based on the slide she evaluated. It is not reasonable to extrapolate macroscopic numbers from one or several microscopic slides. We have discussed this with our pathologist. The reason we reported the weight of the lung metastases was precisely because an attempt at enumeration of metastases from slides had serious statistical limitations. We indicated this in the text. We have included the pathologic enumeration of metastases from slides as provided to us by our blinded pathologist in the Supplemental Data.

There are still multiple minor issues including but not comprehensively; line 194 says 4 week old mice started the diet while line 237 says 6 week old. It is never explained if the mice go off both the treatment and the diet after 5 weeks or do they continue on the diet until death? Too many period on line 201. Line 88 has grams 3 different ways. More than one font is used. 

Thank you for noticing these inconsistencies. The mice were received at approximately 4 weeks of age from Jackson Labs and were started on their diets at approximately 6 weeks, on which they remained until death by euthanasia. This information has been clarified in the text. Multiple “gram” spellings have been condensed to ‘gm’ in all cases. I can’t find extra periods on line 201 or anywhere nearby. My apologies, as I’ve now been over that vicinity many times. I have also double checked the font and it remains Arial 12 throughout the paragraphs, with the exception of the Volume equation (in the Tumor and Survival Measurement section) which was most clearly portrayed using Cambria Math font.

---

## [Decision Letter · Decision Letter 2]

19 Oct 2020

PONE-D-20-11420R2

The Effect of a Ketogenic Diet and Synergy with Rapamycin in a Mouse Model of Breast Cancer.

PLOS ONE

Dear Dr. Fine,

Thank you for submitting your manuscript to PLOS ONE. After careful consideration, we feel that it has merit but does not fully meet PLOS ONE’s publication criteria as it currently stands. Therefore, we invite you to submit a revised version of the manuscript that addresses the points raised during the review process.

Serious objections have been raised in review, and one reviewer has recommended rejection.  We are willing to consider a revised manuscript but these issues need to be addressed. 

We look forward to receiving your revised manuscript.

Kind regards,

Salvatore V Pizzo

Academic Editor

PLOS ONE

Reviewers' comments:

Reviewer's Responses to Questions

**Comments to the Author**

1. If the authors have adequately addressed your comments raised in a previous round of review and you feel that this manuscript is now acceptable for publication, you may indicate that here to bypass the “Comments to the Author” section, enter your conflict of interest statement in the “Confidential to Editor” section, and submit your "Accept" recommendation.

Reviewer #1: (No Response)

Reviewer #2: (No Response)

2. Is the manuscript technically sound, and do the data support the conclusions?

Reviewer #1: Yes

Reviewer #2: No

3. Has the statistical analysis been performed appropriately and rigorously? 

Reviewer #1: Yes

Reviewer #2: I Don't Know

4. Have the authors made all data underlying the findings in their manuscript fully available?

Reviewer #1: Yes

Reviewer #2: Yes

5. Is the manuscript presented in an intelligible fashion and written in standard English?

Reviewer #1: Yes

Reviewer #2: Yes

6. Review Comments to the Author

Reviewer #1: My previous comment appears to NOT have been taken into account; I accept the paper, but would like the authors to change reference 37 into Klement et al. 2020, Med Oncol 37:14 (https://pubmed.ncbi.nlm.nih.gov/31927631/).

Reviewer #2: I do not feel that the authors adequately addressed my concerns from prior reviews. Importantly, my concerns with discussing the limitations of the murine model regarding translation to humans. The murine mouse model they have used is based on a diabetic mouse and this is important for the translation and deserved mention in the discussion. They have added reference 37, which is a review paper that only cites murine model data and they have written that it is a human trial. In both previous reviews I asked them to be cautious about overstating human trial results. Instead of heeding my advice they lectured me on the difference of effectiveness and efficacy. The added pathology quantification data is important but underrepresented by putting it in supplemental.

7. PLOS authors have the option to publish the peer review history of their article (what does this mean?). If published, this will include your full peer review and any attached files.

Reviewer #1: **Yes: **Rainer J. Klement

Reviewer #2: No

---

## [Author Response · Author response to Decision Letter 2]

21 Oct 2020

Reviewer #1: My previous comment appears to NOT have been taken into account; I accept the paper but would like the authors to change reference 37 into Klement et al. 2020, Med Oncol 37:14 (https://pubmed.ncbi.nlm.nih.gov/31927631/).

Response: Our apologies. We have changed the reference as requested.

Reviewer #2: I do not feel that the authors adequately addressed my concerns from prior reviews. Importantly, my concerns with discussing the limitations of the murine model regarding translation to humans. The murine mouse model they have used is based on a diabetic mouse and this is important for the translation and deserved mention in the discussion. They have added reference 37, which is a review paper that only cites murine model data and they have written that it is a human trial. In both previous reviews I asked them to be cautious about overstating human trial results. Instead of heeding my advice they lectured me on the difference of effectiveness and efficacy. The added pathology quantification data is important but underrepresented by putting it in supplemental.

Response: We have changed reference 37 as also requested by Reviewer #1. The new review indeed addresses human studies.

We also have added the reference recommended by Reviewer #2 regarding transgenic mice, particularly the insulin resistance of our mouse strain’s genetic background. Despite our initial skepticism about the added value of this information, it is indeed relevant, and we have added to the discussion accordingly. 

We disagree that the pathology quantification data adds to the fundamental information required in the paper. The numbers of tumors and the size information is too sparse to be statistically significant. The statistically significant data is already within the fundamental figures included. These data are based on macroscopic weight measurements, not a single microscopic slide. But if the editors wish to move this slide into an additional fundamental figure for the paper, we will accept their judgment.

---

## [Editor Report · Decision Letter 3]

9 Nov 2020

The Effect of a Ketogenic Diet and Synergy with Rapamycin in a Mouse Model of Breast Cancer.

PONE-D-20-11420R3

Dear Dr. Fine,

We’re pleased to inform you that your manuscript has been judged scientifically suitable for publication and will be formally accepted for publication once it meets all outstanding technical requirements.

Kind regards,

Salvatore V Pizzo

Academic Editor

PLOS ONE
---

## [Editor Report · Acceptance letter]

16 Nov 2020

PONE-D-20-11420R3 

The Effect of a Ketogenic Diet and Synergy with Rapamycin in a Mouse Model of Breast Cancer. 

Dear Dr. Fine:

I'm pleased to inform you that your manuscript has been deemed suitable for publication in PLOS ONE. Congratulations! Your manuscript is now with our production department. 

Kind regards, 

on behalf of

Dr. Salvatore V Pizzo 

Academic Editor

PLOS ONE